# Spatial–Temporal Evolution Characteristics and Driving Mechanism Analysis of the "Three-Zone Space" in China's Ili River Basin

Zhixian Jiang [1], Meihua Yang [1,*], Li Yang [2], Wenjing Su [3] and Zhuojun Liu [1]

1 Agricultural College, Shihezi University, Shihezi 832003, China
2 Quarantine and Control Section of Harmful Organisms, Forestry and Grassland Work Station of XPCC, Urumqi 830403, China
3 School of Forestry and Landscape Architecture, Xinjiang Agricultural University, Urumqi 830052, China
* Correspondence: ymh_agr@shzu.edu.cn

**Abstract:** The Ili River Basin (IRB) is located in the northwest of China. With its large-scale zone and abundant resources, it is believed to be a "wet island" and a biotic resource storehouse in the dry regions of Eurasia. The IRB has stable ecological conditions and abundant water resources, providing natural conditions for agricultural production and human settlements. With the population increasing and economic development advancing, the competition for land resources is becoming fierce, leading to some ecological problems in this region. Therefore, understanding the spatiotemporal changes and driving mechanisms of the "three-zone space" (TZS) in the IRB is of significant practical importance for promoting sustainable development and optimizing the territorial spatial pattern. This study first analyzes the characteristics and intensity of the TZS changes from 2000 to 2020. Then, it utilizes the optimized parameter Geodetector (OPGD) to analyze the driving mechanisms behind these changes. The results show the following. Firstly, the agricultural space (AS) increased by a total of 837.5 km$^2$, the urban space (US) increased by 519.64 km$^2$, and other ecological space (OES) increased by 1518.83 km$^2$. Green ecological space (GES) decreased by 2875.97 km$^2$. Secondly, intensity analysis indicated that the total TZS change in IRB was 11.07%. At the spatial-type level, the increased intensities of OES, US, and AS were active. In spatial transformation intensity, US and OES tended to transform into AS; AS tended to transform into US; and OES and GES had a mutual transformation tendency. Thirdly, AS converted into US around emerging cities like Khorgas and Cocodala. The conversion towards GES was scattered. The mutual conversion between OES and GES showed spatial distribution consistency, mainly occurring in the Borohoro ranges and the Halik ranges. Lastly, regarding the driving mechanisms, the evolution of US in the IRB was driven by social and economic factors. Location and climate factors accelerated agricultural development, facilitating the transformation of GES and OES into AS. Climate and economic factors played a crucial role in the scale of conversions between OES and GES. The findings can provide a basis for the governance and protection of the IRB, help to form a rational territorial spatial pattern, and offer scientific guidance for sustainable land management.

**Keywords:** spatial–temporal evolution characteristics; "three-zone space"; driving mechanisms; Ili River Basin; optimized parameter Geodetector





## 1. Introduction

The Sustainable Development Goals (SDGs) were released by the United Nations in September 2015 [1]. The SDGs aim to serve as a guide for nations in shaping their planning, policy, and investment strategies, as well as in consistently monitoring and reporting their advancements towards sustainable development from 2016 to 2030 [2]. China actively engages in global environmental and climate governance.

Territorial space is essential for human survival and development; it is a crucial resource for sustainable societal progress and national power [3–5]. China has achieved significant advancements in both its economic and social progress, with accelerated territorial space development in these years. However, issues, such as disorder of spatial development, inefficient resource use, and shrinking ecological space, have emerged [6–8]. Thus, to achieve the SDGs, China has advocated for the establishment of a "Three Zones and Three Lines" framework [9] to create an integrated spatial management system. This framework categorizes territorial space into urban, agricultural, and ecological spaces, each demarcated by the following distinct control lines: the urban development boundary, the boundary for permanent agricultural land, and the ecological protection red line, respectively. The changes in ecological space, agricultural space, and urban space reflect the extent of the influence of human activities [10]. It plays a crucial bridging role between the macro scale (major function-oriented zone [11]) and the micro scale (land use planning [9]). Despite differences in land classification systems and spatial dominant functions, the "three-zone space" (TZS) classification is more suitable for medium to large-scale analyses [12].

The scope of this study is the Ili River Basin (IRB) within China. The Ili River is shared by both China (approximately 624 km long, upstream) and Kazakhstan (approximately 815 km long, downstream) [13,14], with several originated branch rivers including the Tekes River, Kash River, and Gongnaisi River. The IRB, situated in the western part of the Xinjiang Uygur Autonomous Region (XUAR) in Central Asia, is a crucial sensitive ecological environment zone in China [15], and it is known as "Jiangnan beyond the Great Wall". Additionally, it features three border trade ports, namely, Khorgos, Dulata, and Muzhaerte, of which Khorgos stands out as a crucial hub for the New Eurasian Land Bridge, making it the largest land port between China and Kazakhstan [16]. In recent years, the Ili Kazakh Autonomous Prefecture government has introduced and revised the "Ili River Valley Ecological Environment Protection Regulations", formulated the "14th Five-Year Plan for Ecological Environment Protection", and implemented the "Three Lines One Permit" policy [17,18]. These initiatives were designed to regulate the protection of the ecological environment, taking into account factors like industrial scale, spatial distribution, environmental risk mitigation, and ecological carrying capacity, with the goal of achieving a balanced relationship between human activities and the natural world.

With the implementation of the Western Development strategy [19] and socio-economic improvements, the rapid expansion of urban space in the IRB has encroached on ecological and agricultural space, leading to conflicts between development and protection as well as tensions in the relationship between humans and land. To mitigate these issues, it is essential to construct a rationally structured and spatially ordered pattern based on the TZS framework. Specifically, it is crucial to describe the trends and intensities of changes occurring within urban, ecological, and agricultural spaces. By analyzing the distribution, kernel density, and spatial exchange intensity of the TZS from 2000 to 2020, we can determine the scale, intensity, and patterns of the spatial changes in the TZS. This work will help optimize the territorial spatial pattern. The use of a space pattern transfer matrix [20], kernel density estimation [21], and TZS dynamic degree [11] methods is suitable for achieving this goal. Current research on the IRB primarily focuses on the sources of soil and potential distribution mechanisms [22–24], the distribution and impact mechanisms of diseases [25], and areas such as hydrology [26], the ecological environment [27], and climate [28,29]. There is limited research on land use [30,31] and a lack of studies on the evolution processes and impact mechanisms of the TZS in the IRB from the perspective of territorial spatial planning. The transition matrix is a fundamental method in the analysis of land change [32]. To explain the changes in the TZS, one can utilize the land use transition matrix method to obtain accurate information about the potential processes that drive the changes. Intensity analysis is a hierarchical analysis method used to analyze land use change in terms of its interval, category, and transition level [33–35]. This study utilizes intensity analysis to assess the magnitude and intensity of total losses and gains for each

spatial type at the category level between 2000 and 2020. At the transition level, it analyzes the size and intensity of specific spatial types transitioning to other categories between 2000 and 2020. Regarding driving mechanisms, the optimized parameter Geodetector tool (OPGD) [36] has seen extensive use across diverse domains, such as assessing ecological risk factors, examining the spatiotemporal dynamics of land use and land cover changes, and analyzing spatial variation characteristics [37–40]. This study utilizes OPGD to explore the impact of economy, society, geography, location, and climate factors on the evolution of the TZS, aiming to provide a more comprehensive understanding of the spatiotemporal evolution characteristics and mechanisms of TZS in the IRB.

The IRB is a key hub of the Silk Road Economic Belt and a strategic ecological security barrier in Northwest China, which is considered a "wet island" and a biotic resource storehouse in the dry regions of Eurasia. Within the framework of the TZS, this research quantitatively analyzes spatial and temporal cross-transformation characteristics and detects the driving mechanisms behind the conversion of the TZS in the IRB from 2000 to 2020. Furthermore, it fills a research gap in the long-term analysis of the TZS within the context of territorial spatial planning in the IRB. The aim is to provide a basis for strengthening the governance and protection of the region, accelerate the formation of a territorial spatial pattern, and promote the achievement of SDGs.

## 2. Materials and Methods

### 2.1. Study Area and Data Sources

#### 2.1.1. Study Area

The IRB in China (42°14′16″–44°53′30″ N, 80°09′42″–84°56′50″ E) is located in the interior of Central Asia in the western part of XUAR (Figure 1), with an area of approximately 55,339.16 km$^2$, including 3 cities and 8 counties. By the end of 2020, the total population of the area was 2,619,200 with a population density of 47 people/km$^2$. The study area encompasses the following three border ports: Khorgos, Dulata, and Muzhaerte. The IRB is an area that is a sensitive ecological environment. The "Ili Prefecture Territorial Spatial Planning 2021–2035" positions the IRB as an important hub of the Silk Road Economic Belt, a world-class tourist destination, and a strategic ecological security barrier in Northwest China.

The IBR features complex terrain, with terrain that is higher in the east and lower in the west. It is narrow in the east and wide in the west, resembling a trumpet opening to the west. The Ili River originates from three main feeder streams (the Kash, Gongnaisi, and Tekes Rivers), which descend from the Tian Shan Mountains. The river is surrounded by the Borohoro and Dzungarian Alatau ranges to the north and the Halik and Ketpen ranges to the south. It flows westward for 1,439 km from its point of origin, exiting China's XUAR and extending into Kazakhstan [41], with a mean annual discharge of about 480 m$^3$/s [14]. The IRB, which has a typical temperate continental climate characterized by a mean annual temperature of 7 °C [42] and a mean annual precipitation ranging from 200 to 1000 mm, is the wettest region in XAUR [43]. It features a relatively humid climate, well-developed vegetation, and pasture, making it a typical livestock farming base in China [44].

#### 2.1.2. Data Sources

This study focuses on the TZS of the IRB, analyzing its distribution, scale, spatial transformation, and driving mechanisms. To ensure data consistency, the administrative units of 2020 were used as the reference. The necessary data are listed in Table 1.

### 2.2. Methods

#### 2.2.1. Spatial Classification

This article used ArcGIS technology to crop GlobeLand30 data from 2000 to 2020. The overall accuracy of the GlobeLand30 is 83.58% in XUAR, with a kappa coefficient of 0.717 [47]. Based on the GlobeLand30 classification of land use types, the nine distinct categories within the region were subsequently grouped into three overarching spaces as fol-

lows: "ecological space, agricultural space, and urban space" (Table 2). Urban space (US) primarily consists of artificial surfaces, while agricultural space (AS) is predominantly cultivated land. The special characteristics of bareland in the basin ecological space (large-scale and concentrated bareland; obvious cross-conversion with forest, shrubland, and other land uses; and dominant functions and ecological services are different from forest and other land uses) necessitated its classification into two subcategories as follows: "green ecological space" (GES) and "other ecological space" (OES).

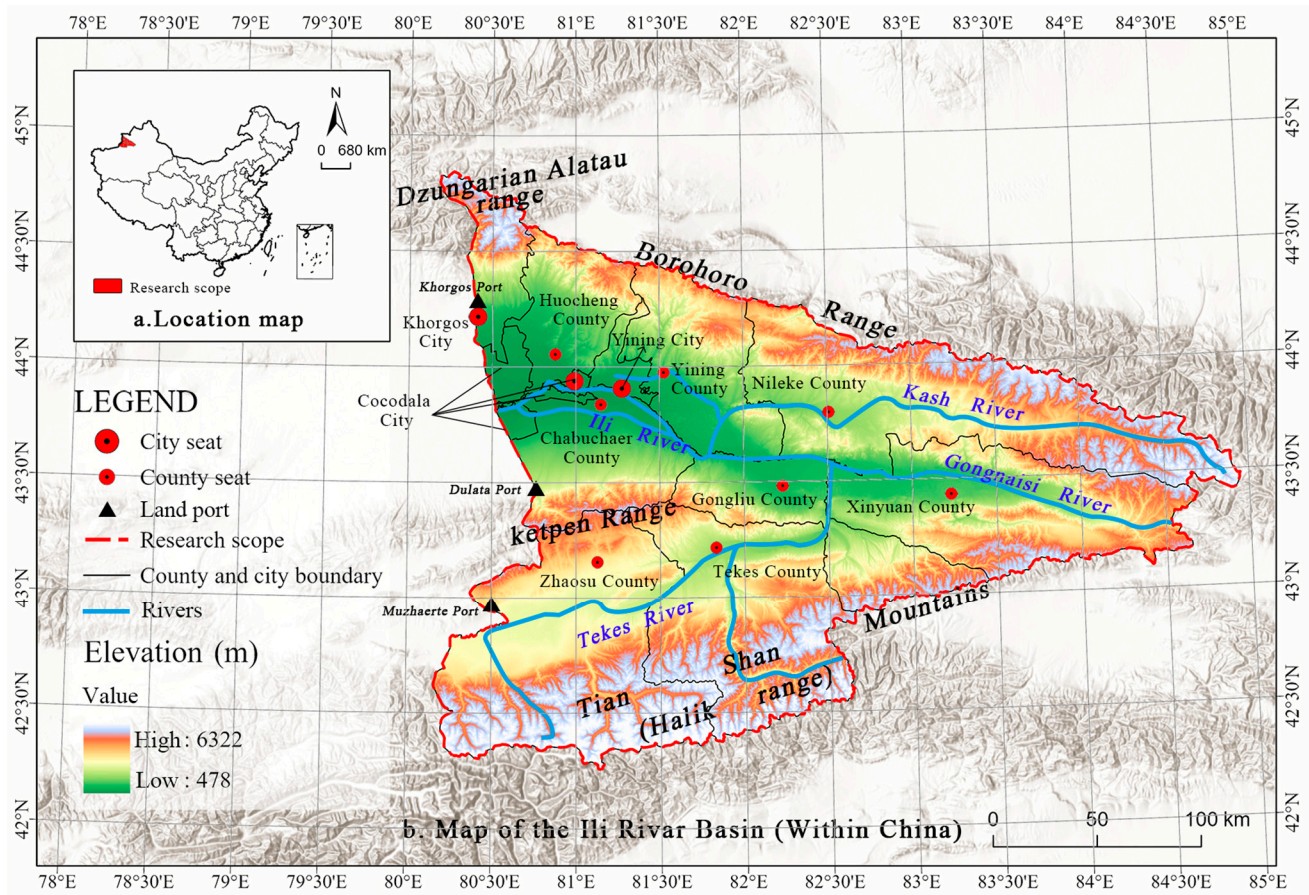

**Figure 1.** Geographical map of the study area.

**Table 1.** Data sources.

| Data Classification | Data Name | Variable | Data Source |
|---|---|---|---|
| Land use basic data | 30 m Global Surface Coverage Data GlobeLand30 (2000–2020) | | https://www.webmap.cn/commres.do?method=globeDetails&type=brief (accessed on 3 June 2023) https://zenodo.org/records/8385299 (alternative link) (accessed on 10 September 2023) [45]. |
| | National 1:1,000,000 Basic Geographical Information Dataset (2021) | X12, X13 X14, X15 | https://www.webmap.cn/ (accessed on 12 May 2023). |
| | Natural geographic data | X8, X9, X10, X11 | ASTER GDEM 30M (https://www.gscloud.cn/ (accessed on 12 May 2023)). Spatial distribution of soil types in China (https://www.resdc.cn/ (accessed on 12 May 2023)). |

| Data Classification | Data Name | Variable | Data Source |
|---|---|---|---|
| Driver mechanism data | Annual economic data | X1, X2 X3, X4 | China County Statistical Yearbook (2021). Statistical Yearbook of the XPCC (2021). Annual dataset of nighttime lighting in China (http://www.resdc.cn (accessed on 14 May 2023)) [46]. |
| | Annual social data | X5, X6, X7 | Seventh Population Census Data (stats.gov.cn (accessed on 4 June 2023)). ORNL LandScan Viewer-Oak Ridge National Laboratory (accessed on 5 June 2023) |
| | Climatic data | X16, X17 X18 | National Earth System Science Data Center, National Science, and Technology Infrastructure of China (https://www.geodata.cn (accessed on 10 June 2023)). National Cryosphere Desert Data Center (http://www.ncdc.ac.cn (accessed on 12 July 2023)). |

Note: X1–X18 correspond to various influencing factors in Section 2.2.8.

**Table 2.** The classification scheme of "three-zone space" in the Ili River Basin.

| Spatial Type | | GlobeLand30 Land-Use Classification System | | Classification Basis |
|---|---|---|---|---|
| Primary Space | Secondary Space | Primary Land Use Classification | Secondary Land Use Classification | |
| Urban space (US) | – | Artificial surfaces | Urban residential land, rural residential land, industrial and mining land, transportation facility land, etc. | Carry the town's development |
| Agricultural space (AS) | – | Cultivated land | Paddy land, irrigated drylands, rainfed drylands, vegetable plots, pasture land, greenhouses, economic tree forests, economic shrub forests. | Ensure food security |
| Ecological space | Other ecological space (OES) | Bareland | Deserts, sand, gravel, bare rock, salt marshes, etc. | Need for ecological restoration |
| | Green ecological space (GES) | Forest land | Land with tree cover and more than 30% canopy cover, open forest land with 10% to 30% cover. | Provision of ecological services such as soil conservation, water conservation, climate regulation, and biodiversity conservation |
| | | Grassland | Grasslands, meadow savannas, desert grasslands, artificial grasslands. | |
| | | Shrubland | Mountain scrub, deciduous and evergreen scrub, and desert scrub. | |
| | | Wetland | Bogs, riverine floodplains, forested/shrubland wetlands, peat bogs, salt marshes, etc. | |
| | | Water bodies | Rivers, lakes, reservoirs, ponds, offshore, etc. | |
| | | Permanent snow and ice | Permanent snow in alpine areas and glaciers. | |

Therefore, this paper established the TZS classification system in the IRB by taking land dominant functions as the starting point, combining relevant studies with the actual situation of the basin. This study analyzed the evolutionary characteristics of US, AS, OES, and GES.

### 2.2.2. "Three-Zone Space" Dynamic Degree

The dynamic degree serves as a measure to quantify both the rate of transformation and the extent of change that various geographical functional areas undergo within a specified time frame in research on land use change [48,49].

$$K = \frac{(U_b - U_a)}{U_a} \times \frac{1}{T} \times 100\%$$ (1)

where $K$ represents the dynamic degree, or the rate of change, of a particular spatial type over a defined time period. $U_a$ signifies the initial area occupied by this spatial type at the start of the period, whereas $U_b$ denotes its area at the conclusion of the period. $T$ stands for the length of this time period.

### 2.2.3. Kernel Density Estimation

Kernel density estimation can help us understand the spatial distribution patterns of data points [50]. This study utilized ArcGIS 10.8 software to analyze the spatial distribution characteristics of the transformed area within the TZS. After comparing kernel density analysis maps under various search radii, a 10 km search radius was adopted, as it better depicted the spatial distribution. The formula is as follows:

$$F_n(x) = \frac{1}{nh} \sum_{i=1}^{n} k\left(\frac{x - x_i}{h}\right)$$ (2)

where $F_n(x)$ serves as an approximation for the kernel density value for the search radius, denoted as $h$ ($h > 0$). $n$ signifies the total count of samples that represent spatial transitions within the studied area. The $k$ function quantifies the distance between any given point element $x$ and a reference point $xi$.

### 2.2.4. Space Pattern Transfer Matrix

A land use transfer matrix can visualize the overall change in land use types [51]. This study employed a land use transfer matrix as a tool for quantitatively describing the transformation of TZS. In Table 3, $i$ = 1, 2, … $J$ represents the space type at the initial time point and $j$ = 1, 2, … $J$ represents the space type at the final time point, where $J$ is the number of space types. $T$ is the number of time points, $Y_t$ represents the year at time point $t$, and $t$ represents the index for the initial time point of interval $[Y_t, Y_{t+1}]$, where $t$ ranges from 1 to $T - 1$. $C_{tij}$ represents the area converted from space type $i$ at time $Y_t$ to space type $j$ at time $Y_{t+1}$. $C_{tii}$ represents the area of space type $i$ that remains unchanged. $\sum_{j=1}^{J} C_{tij}$ represents the total area of space type $i$ at the initial time point. $\sum_{i=1}^{J} C_{tij}$ represents the total area of space type $j$ at the final time point. $\left(\sum_{j=1}^{J} C_{tij}\right) - C_{tii}$ represents the loss in the area of space type $i$ for time interval $[Y_t, Y_{t+1}]$; $\left(\sum_{i=1}^{J} C_{tij}\right) - C_{tjj}$ represents the gain in the area of land use type $j$ for time interval $[Y_t, Y_{t+1}]$.

**Table 3.** Space pattern transfer matrix from the initial time to the final time.

| | | Categories at Final Time ($t$ + 1) | | | Initial Total ($Y_t$) | Gross Loss |
|---|---|---|---|---|---|---|
| | | $j$ = 1 | $j$ = 2 | ... | $j$ = $J$ | | |
| Categories at Initial Time ($t$) | $i$ = 1 | $C_{t11}$ | $C_{t12}$ | ... | $C_{t1J}$ | $\sum_{j=1}^{J} C_{t1j}$ | $\left(\sum_{j=1}^{J} C_{t1j}\right) - C_{t11}$ |
| | $i$ = 2 | $C_{t21}$ | $C_{t22}$ | ... | $C_{t2J}$ | $\sum_{j=1}^{J} C_{t2j}$ | $\left(\sum_{j=1}^{J} C_{t2j}\right) - C_{t22}$ |
| | ⋮ | ... | ... | ... | ... | ... | ... |
| | $i$ = $J$ | $C_{tJ1}$ | $C_{tJ2}$ | ... | $C_{tJJ}$ | $\sum_{j=1}^{J} C_{tJj}$ | $\left(\sum_{j=1}^{J} C_{tJj}\right) - C_{tJJ}$ |
| Final Total ($Y_{t+1}$) | | $\sum_{i=1}^{J} C_{ti1}$ | $\sum_{i=1}^{J} C_{ti2}$ | ... | $\sum_{i=1}^{J} C_{tiJ}$ | $\sum_{i=1}^{J} \sum_{j=1}^{J} C_{tij}$ | $\sum_{i=1}^{J}\left(\left(\sum_{j=1}^{J} C_{tij}\right) - C_{tii}\right)$ |
| Gross Gain | | $\left(\sum_{i=1}^{J} C_{ti1}\right) - C_{t11}$ | $\left(\sum_{i=1}^{J} C_{ti2}\right) - C_{t22}$ | ... | $\left(\sum_{i=1}^{J} C_{tiJ}\right) - C_{tJJ}$ | $\sum_{j=1}^{J}\left(\left(\sum_{i=1}^{J} C_{tij}\right) - C_{tjj}\right)$ | |

### 2.2.5. Change Budget

Based on the spatial change transfer matrix, we can further calculate the spatial change budget [52]. Equation (3) calculates the total change by summing all the entries within the matrix and then subtracting the diagonal entries, which represent persistence. The total change encompasses both quantity change and allocation change.

$$Total\ change = S = \frac{\sum_{j=1}^{J}\left\{\left(\sum_{i=1}^{J} C_{tij}\right) - C_{tij}\right\}}{\sum_{j=1}^{J}\sum_{i=1}^{J} C_{tij}} \times 100\% \tag{3}$$

$$Quantity\ change = \frac{\sum_{j=1}^{J}\left\{MAXIMUM\left[0, \sum_{i=1}^{J}\left(C_{tij} - C_{tji}\right)\right]\right\}}{\sum_{j=1}^{J}\sum_{i=1}^{J} C_{tij}} \times 100\% \tag{4}$$

$$Allocation\ change = Total\ change - Quantity\ change \tag{5}$$

Equation (4) gives the quantity change. Categories with a positive net change are selected by the *MAXIMUM* function, followed by a summation over *j* to accumulate the positive net changes. Equation (5) computes the allocation change, which represents the percentage of total change minus the percentage of quantity change.

### 2.2.6. Intensity Analysis

The intensity analysis method, grounded in a transfer matrix, constitutes a mathematical framework that compares a uniform intensity with the intensities of temporal variations observed among distinct categories [33,53]. Although this study utilized only a single time interval from 2000 to 2020, we could still employ the interval level of intensity analysis to calculate the annual change during the interval $[Y_t, Y_{t+1}]$, which serves as a foundation for subsequent category level analysis. In Equation (6), $S_t$ represents the observed annual change intensity during interval $[Y_t, Y_{t+1}]$.

$$S_t = \frac{\left\{\sum_{j=1}^{J}\left[\left(\sum_{i=1}^{J} C_{tij}\right) - C_{tij}\right]\right\} / \left[\sum_{j=1}^{J}\sum_{i=1}^{J} C_{tij}\right]}{Y_{t+1} - Y_t} \times 100\% \tag{6}$$

This study adopted this method at two levels (category and transition) to analyze the intensity characteristics of changes in the TZS.

$$G_{tj} = \frac{\left[\left(\sum_{i=1}^{J} C_{tij}\right) - C_{tjj}\right] / \left(\sum_{i=1}^{J} C_{tij}\right)}{Y_{t+1} - Y_t} \times 100\% \tag{7}$$

$$L_{ti} = \frac{\left[\left(\sum_{j=1}^{J} C_{tij}\right) - C_{tii}\right] / \left(\sum_{j=1}^{J} C_{ij}\right)}{Y_{t+1} - Y_t} \times 100\% \tag{8}$$

Equations (7) and (8) show the intensity analysis of the category level, where $G_{tj}$ represents the annual average increase intensity and $L_{ti}$ represents the annual average decrease intensity. By comparing the values of $G_{tj}$, $L_{ti}$, and $S_t$ (from Equation (6)), it can be determined whether the increase or decrease in a space type is active or dormant. If $G_{tj} > S_t$, the increase in category *j* is judged as active; if $G_{tj} < S_t$, the increase in category *j* is judged as dormant. The same applies to the judgment of space type decrease.

$$R_{tin} = \frac{(C_{tin}) / \left(\sum_{j=1}^{J} C_{tij}\right)}{Y_{t+1} - Y_t} \times 100\% \tag{9}$$

$$W_{tn} = \frac{\left[\left(\sum_{i=1}^{J} C_{tin}\right) - C_{tnn}\right] / \sum_{j=1}^{J}\left\{\left(\sum_{i=1}^{J} C_{ij}\right) - C_{nj}\right\}}{Y_{t+1} - Y_t} \times 100\% \tag{10}$$

Equations (9) and (10) show the intensity analysis of the transition level, where $R_{tin}$ represents the intensity of the transformation from other space types to this specific space type and $W_{tn}$ represents the uniform transition intensity of other space types converting to this space type. If space types $n$ were to gain with the same intensity from all non-$i$ space types, then $R_{tin}$ would equal $W_{tn}$. If $R_{tin} > W_{tn}$, it is judged that space type $i$ tends to convert to space type $n$; if $R_{tin} < W_{tn}$, it is judged that space type $i$ avoids converting to space type $n$.

### 2.2.7. Optimized Parameter Geodetector

The factor detection and interaction detection of the optimized parameter Geodetector (OPGD) [36] was employed to analyze the impact of various factors on the transformation of the TZS.

In practical spatial analysis, continuous variables should be discretized with optimal parameters before modeling. The efficacy of discretization in classification can be assessed through the statistic $Q$ of a geographic detector, where a higher $Q$ value indicates better partitioning. Thus, we employed the GD package in the R language, utilizing methods such as natural breaks, quantile breaks, and others, with the number of classifications ranging from 3 to 8, and selected the parameter combination yielding the highest $Q$ value as the parameter for geographic detector analysis.

$$Q_v = 1 - \frac{\sum_{j=1}^{M} N_{v,j} \sigma_{v,j}^2}{N_v \sigma_v^2} \tag{11}$$

Equation (11) show the explanatory power of factors for the transformation of various types of spaces, with the range of $Q$-statistic values being [0, 1]. The larger the $Q$, the stronger the explanatory power of the factor for the specific type of spatial transformation. $j = 1, \ldots, M$ represents the strata (or sub-regions) of the explanatory variable $v$; $N_{v,j}$ and $N_v$ represent the number of units in the $j$-th sub-region of variable $v$ and the whole area, respectively. $\sigma_{v,j}^2$ and $\sigma_v^2$ are the variances in the $j$-th sub-region of variable $v$ and the whole area, respectively.

The impacts of two spatial variables were explained by their interactions, which are illustrated by the five interaction types shown in Table 4 [54,55].

**Table 4.** Types of interaction between two covariates.

| Geographical Interaction Relationship | Interaction |
|---|---|
| $Q_{u \cap v} < \min(Q_u, Q_v)$ [1] | Nonlinear weaken: Impacts of single variables are nonlinearly weakened by the interaction between two variables. |
| $\min(Q_u, Q_v) \leq Q_{u \cap v} \leq \max(Q_u, Q_v)$ | Univariable weaken: Impacts of single variables are univariable weakened by the interaction. |
| $\max(Q_u, Q_v) < Q_{u \cap v} < (Q_u + Q_v)$ | Bi-variable enhance: Impact of single variables are bi-variable enhanced by the interaction. |
| $Q_{u \cap v} = (Q_u + Q_v)$ | Independent: Impacts of variables are independent. |
| $Q_{u \cap v} > (Q_u + Q_v)$ | Nonlinear enhance: Impacts of variables are nonlinearly enhanced. |

[1] $Q_u$ is the Q-statistic of the variable $u$, $Q_v$ is the Q-statistic of the variable $v$, and $Q_{u \cap v}$ is the interaction Q-statistic between the variables $u$ and $v$.

### 2.2.8. Description of Variables

Considering local circumstances and the factors' quantifiabilities and availabilities, we selected 18 indicators from the perspectives of economy, society, geography, location, and climate (Table 5). The selected independent variables were stratified using optimal parameter selection, converting them from numerical variables to categorical variables. Then, using the OPGD, detection was conducted for the 12 spatial types in the IRB from 2000 to 2020.

**Table 5.** Description of variables and indicators.

| Dimensions of Influencing Factors | Name of Independent Variable | Processing Method |
|---|---|---|
| Economy-related | Output value of primary production (X1)<br>Nighttime light index (X2)<br>Local general budget revenue (X3)<br>Local general budget expenditure (X4) | Statistical yearbook acquisition<br>ArcGIS raster statistics<br>Statistical yearbook acquisition<br>Statistical yearbook acquisition |
| Social-related | Resident population (X5)<br><br>Urbanization rate (X6)<br><br>Population density (X7) | The seventh national population census<br>Urban registered population/total registered population<br>ArcGIS raster statistics |
| Geography-related | Elevation (X8)<br>Slope (X9)<br>Slope aspect (X10)<br>Soil type (X11) | ArcGIS raster statistics<br>ArcGIS raster statistics<br>ArcGIS raster statistics<br>ArcGIS raster statistics |
| Location condition | Average distance to city and county seats (X12)<br>Average distance to rivers (X13)<br>Average distance to railways (X14)<br>Average distance to roads (X15) | ArcGIS Euclidean distance analysis<br>ArcGIS Euclidean distance analysis<br>ArcGIS Euclidean distance analysis<br>ArcGIS Euclidean distance analysis |
| Climate-related | Average annual temperature (X16)<br>Average annual precipitation (X17)<br>Snow depth (X18) | ArcGIS raster statistics<br>ArcGIS raster statistics<br>ArcGIS raster statistics |

## 3. Results

### 3.1. Characteristics of Spatial–Temporal Evolution of Land Use Structure

From 2000 to 2020, significant changes were observed in the land use distribution in the IRB (Figure 2 and Table 6). The cultivated land increased by 837.5 km$^2$ over 20 years, with a gradual decline in the first decade and substantial growth in the second. The artificial surfaces expanded by 519.64 km$^2$, with the majority of the increase concentrated in the cities of Khorgos, Cocodala, and Yining, while other counties and cities also experienced different levels of growth.

The reduction of 1469.2 km$^2$ occurred in the grassland, representing a 4.46% decrease. The decline in grassland primarily occurred from 2000 to 2010. During the following decade, the reduction in grassland area was alleviated. Shrubland decreased by 11.64 km$^2$ over a period of 20 years, while bareland increased by 1518.20 km$^2$. Bareland was primarily found in the southern region of the Tian Shan Mountains, the Borohoro range, and Dzungarian Alatau range, intersecting with glacier and permanent snow land as well as forested land. This trend will increase the ecological vulnerability of the IRB. The most substantial growth in land use area was observed in wetlands, with their size increasing from 8.94 km$^2$ in 2000 to 300.60 km$^2$ in 2020, representing a growth rate of 3264.27%. They were primarily located along the riverbanks, and during the same period, the water bodies of the IRB also increased to 381.65 km$^2$, with a growth rate of 112.72%. The expansion of wetland areas was closely linked to the increase in water area. Glaciers and permanent snow land saw the most significant decrease in area, declining by 1996.83 km$^2$, which was attributed to global warming and the overall temperature rise in the IRB [15]. Meanwhile, the melting of glaciers and permanent snowfields accelerated the formation of bareland.

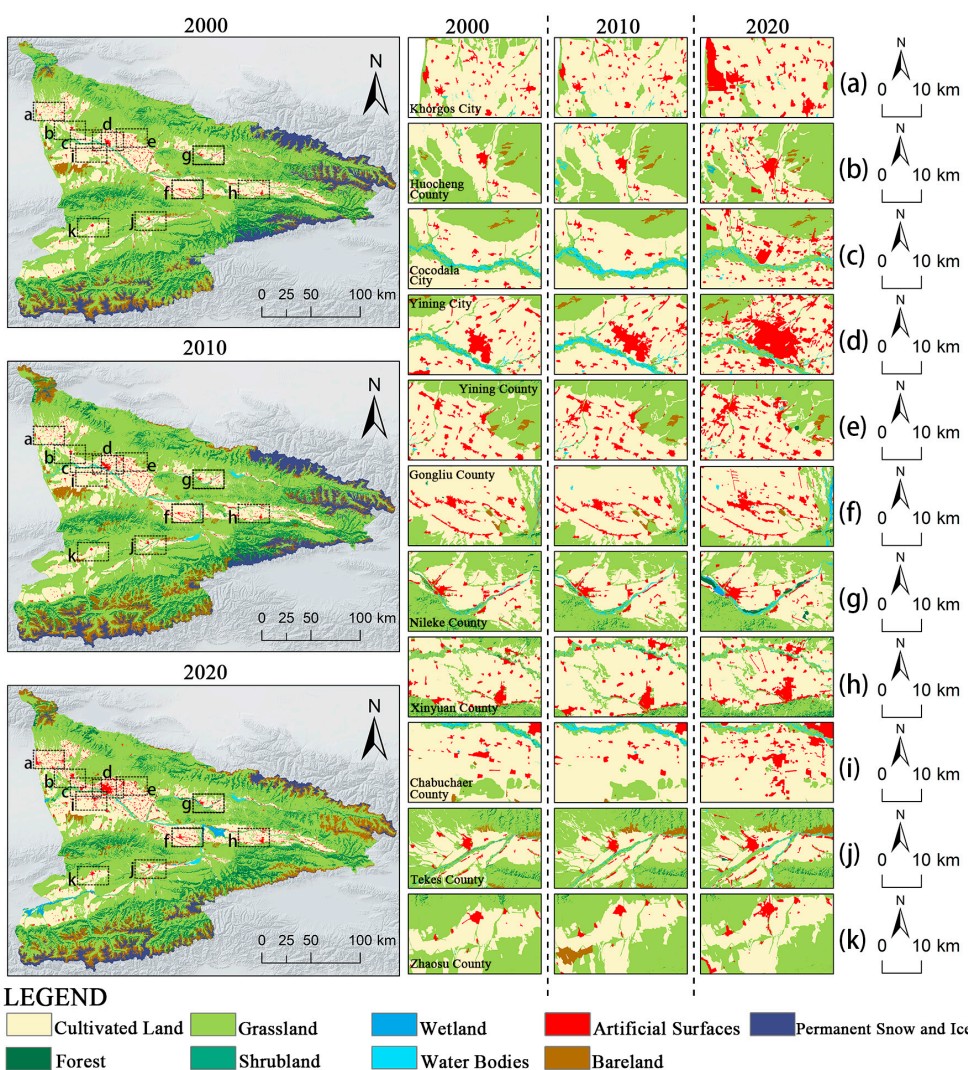

**Figure 2.** Land use classification maps of the IRB in 2000, 2010, and 2020. Note: (**a**) Khorgos City, (**b**) Huocheng County, (**c**) Cocodala City, (**d**) Yining City, (**e**) Yining County, (**f**) Gongliu County, (**g**) Nileke County, (**h**) Xinyuan County, (**i**) Chabuchaer County, (**j**) Tekes County, (**k**) Zhaosu County.

**Table 6.** Land use classification data of the Ili River Basin in 2000, 2010, and 2020.

| Type of Land Use | 2000 Land Area (km²) | 2000 Proportion of Total Area (%) | 2010 Land Area (km²) | 2010 Proportion of Total Area (%) | 2020 Land Area (km²) | 2020 Proportion of Total Area (%) | Rate of Change in Total Area, 2000–2010 (%) | Rate of Change in Total Area, 2010–2020 (%) | Rate of Change in Total Area, 2000–2020 (%) |
|---|---|---|---|---|---|---|---|---|---|
| Cultivated land | 9123.57 | 16.49 | 9107.26 | 16.46 | 9961.07 | 18 | −0.18 | 9.38 | 9.18 |
| Forest | 5172.58 | 9.35 | 5312.44 | 9.6 | 5280.37 | 9.54 | 2.7 | −0.6 | 2.08 |
| Grassland | 32,907.95 | 59.47 | 30,962.83 | 55.95 | 31,438.75 | 56.81 | −5.91 | 1.54 | −4.46 |
| Shrubland | 74.64 | 0.13 | 73.11 | 0.13 | 63 | 0.11 | −2.05 | −13.83 | −15.59 |
| Wetland | 8.94 | 0.02 | 12.2 | 0.02 | 300.6 | 0.54 | 36.58 | 2363.16 | 3264.27 |
| Water bodies | 179.41 | 0.32 | 337.89 | 0.61 | 381.65 | 0.69 | 88.33 | 12.95 | 112.72 |
| Artificial surfaces | 599.47 | 1.08 | 627.34 | 1.13 | 1119.11 | 2.02 | 4.65 | 78.39 | 86.68 |
| Bareland | 2935.2 | 5.3 | 4621.65 | 8.35 | 4454.02 | 8.05 | 57.46 | −3.63 | 51.75 |
| Permanent snow and ice | 4337.42 | 7.84 | 4284.44 | 7.74 | 2340.59 | 4.23 | −1.22 | −45.37 | −46.04 |
| Total | 55,339.16 | 100 | 55,339.16 | 100 | 55,339.16 | 100 | | | |

### 3.2. Spatial–Temporal Evolution Characteristics of the "Three-Zone Space" in the Ili River Basin

From 2000 to 2020, notable alterations took place in the urban spatial configuration of the IRB (Figure 3 and Table 7). In 2020, US reached 1119.11 km², accounting for 2.02% of the total study area. Over the course of 20 years, US increased by a total of 519.64 km², representing a growth rate of 86.68%. This increase was primarily concentrated in newly established cities, such as Khorgos City and Cocodala City, and the surrounding areas of Yining City (Figure 3a–d). The evolution rate of US shifted from stable growth in the first decade to rapid expansion in the second decade (dynamic degree: 0.46 to 7.84%). In terms of spatial distribution, US was primarily distributed along the valley lowlands. Taking Yining City as an example, urbanization led to gradual outward expansion of its US, resulting in the formation of an urban–rural interface with adjacent agricultural areas. From 2000 to 2020, the overall scale of AS underwent a process of initial decline followed by growth (dynamic degree: −0.02 to 0.94%). From 2000 to 2010, there was a slight decrease in AS, amounting to a reduction of 16.31 km². Over the next decade, there was a stable increase, with a growth scale of 853.81 km². In 2020, AS reached 9961.07 km², accounting for 18% of the total study area. Over the course of 20 years, the AS increased by a total of 837.5 km², representing a growth rate of 9.18%. Notably, the expansion of AS was particularly evident in Chabuchaer County (Figure 3i).

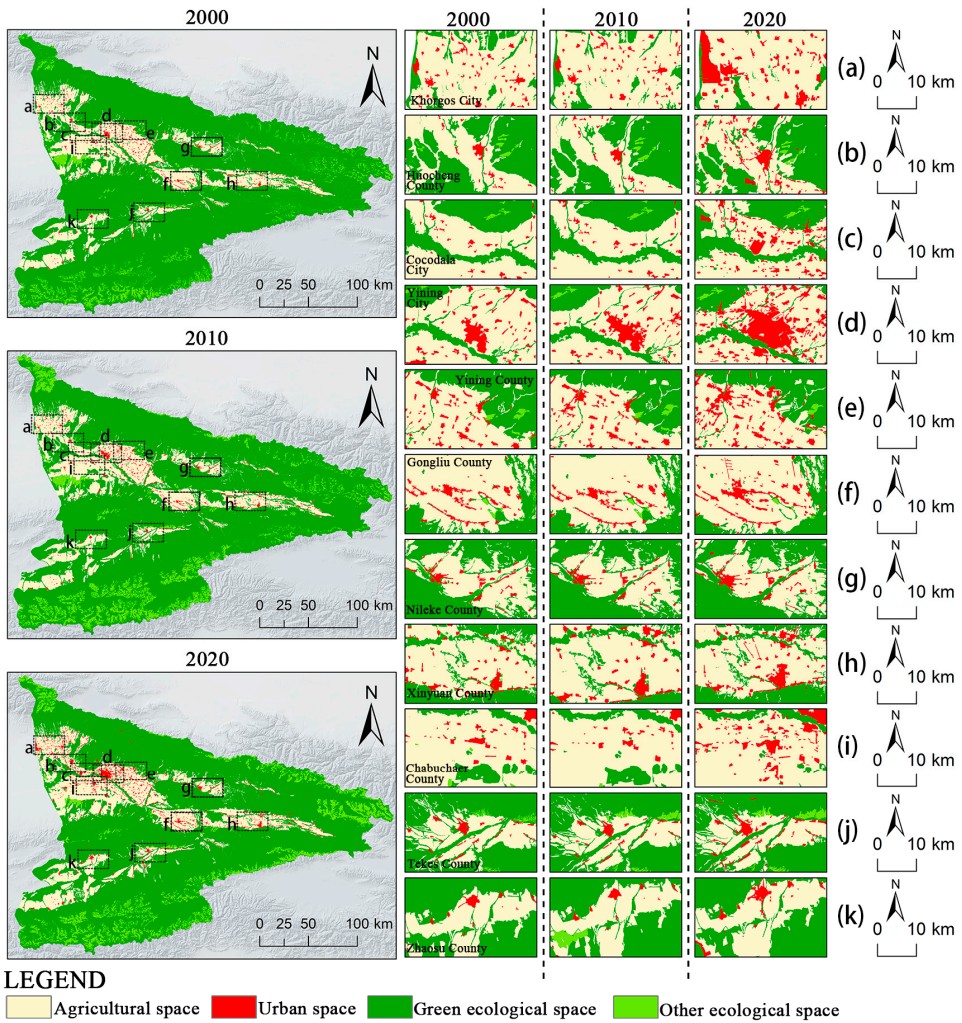

**Figure 3.** The three-zone space evolution in the Ili River Basin in 2000, 2010, and 2020. Note: (**a**) Khorgos City, (**b**) Huocheng County, (**c**) Cocodala City, (**d**) Yining City, (**e**) Yining County, (**f**) Gongliu County, (**g**) Nileke County, (**h**) Xinyuan County, (**i**) Chabuchaer County, (**j**) Tekes County, (**k**) Zhaosu County.

There was a decrease in the scale of ecological space, with the area decreasing from 45,616.13 km² in 2000 to 44,258.98 km² in 2020, representing a reduction rate of 2.98%. Furthermore, from a secondary classification standpoint, there was a downward trend in GES. The net decrease in GES from 2000 to 2020 was 2875.97 km², with a dynamic degree of −0.67%. For OES, the net increase from 2000 to 2010 was 1686.45 km², with a dynamic degree of 5.75%, and the net decrease from 2010 to 2020 was 167.93 km², with a dynamic degree of −0.36%. These spaces were mostly situated in the mountainous areas surrounding the IRB.

**Table 7.** Spatial changes and dynamic degree of the "Three-zone space" in the Ili River Basin in 2000, 2010, and 2020.

| | | AS | US | ES | | Total |
| | | | | GES | OES | |
|---|---|---|---|---|---|---|
| 2000 | Area (km²) | 9123.57 | 599.47 | 42,680.93 | 2935.2 | 55,339.16 |
| | Proportion of total area (%) | 16.5 | 1.08 | 77.14 | 5.28 | 100 |
| 2010 | Area (km²) | 9107.26 | 627.34 | 40,982.91 | 4621.65 | 55,339.16 |
| | Proportion of total area (%) | 16.46 | 1.13 | 74.04 | 8.37 | 100 |
| 2020 | Area (km²) | 9961.07 | 1119.11 | 39,804.96 | 4454.02 | 55,339.16 |
| | Proportion of total area (%) | 18 | 2.02 | 71.92 | 8.06 | 100 |
| 2000–2010 | Area change (km²) | −16.31 | 27.87 | −1698.02 | 1686.45 | — |
| | Growth rate (%) | −0.18 | 4.65 | −3.98 | 57.46 | — |
| | Dynamic degree (%) | −0.02 | 0.46 | −0.4 | 5.75 | — |
| 2010–2020 | Area change (km²) | 853.81 | 491.77 | −1177.95 | −167.93 | — |
| | Growth rate (%) | 9.38 | 78.39 | −2.87 | −3.63 | — |
| | Dynamic degree (%) | 0.94 | 7.84 | −0.29 | −0.36 | — |
| 2000–2020 | Area change (km²) | 837.5 | 519.64 | −2875.97 | 1518.82 | — |
| | Growth rate (%) | 9.18 | 86.68 | −6.74 | 51.75 | — |
| | Dynamic degree (%) | 0.92 | 8.67 | −0.67 | 5.17 | — |

Note: AS, agricultural space; US, urban space; GES, green ecological space; OES, other ecological space.

### 3.3. Spatial–Temporal Cross-Transformation Characteristics of the "Three-Zone Space"

#### 3.3.1. Change Budget for "Three Zone Space"

According to Table 8, from 2000 to 2020, there was a change in the spatial area of the TZS within the study area. Specifically, US increased by 584 km², and the increase in OES was the highest, reaching 2799.35 km². The main reduction was the decrease in GES, with a reduction of 4165.98 km². The reduction in US was the least, at only 64.36 km².

**Table 8.** Transfer matrix of the three-zone space, 2000–2020 (km²).

| Projects | | 2020 | | | | Total 2000 | Gross Loss |
| | | AS | US | GES | OES | | |
|---|---|---|---|---|---|---|---|
| 2000 | AS | 8506.36 | 408.86 | 206.43 | 1.92 | 9123.57 | 617.21 |
| | US | 52.25 | 535.10 | 12.05 | 0.05 | 599.47 | 64.36 |
| | GES | 1209.95 | 158.66 | 38,514.95 | 2797.38 | 42,680.93 | 4165.98 |
| | OES | 192.51 | 16.48 | 1071.53 | 1654.67 | 2935.20 | 1280.52 |
| Total 2020 | | 9961.07 | 1119.11 | 39,804.96 | 4454.02 | 55,339.16 | 6128.07 |
| Gross Gain | | 1454.71 | 584 | 1290.01 | 2799.35 | 6128.07 | |

Note: AS, agricultural space; US, urban space; GES, green ecological space; OES, other ecological space.

Figure 4 shows that the total spatial change in the TZS of the IRB from 2000 to 2020 was 11.07%, comprising both quantity and allocation changes, with quantity change accounting for 5.20% of the total.

Figure 5 illustrates the percent gain, persistence, and loss in the total area from 2000 to 2020. By calculating the sum of persistence and gain for each spatial type in the diagram,

one can obtain the scale of the corresponding spatial type in 2020. Similarly, by calculating the sum of persistence and loss, one can obtain the scale of the corresponding spatial type in 2000. The percentage of GES had the greatest loss, whereas US had the least. In contrast, the percentage of OES had the largest gain.

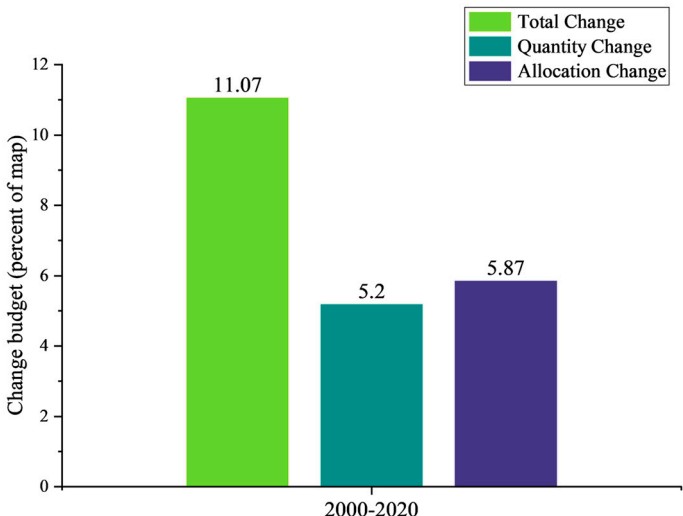

**Figure 4.** Total change separated into quantity and allocation.

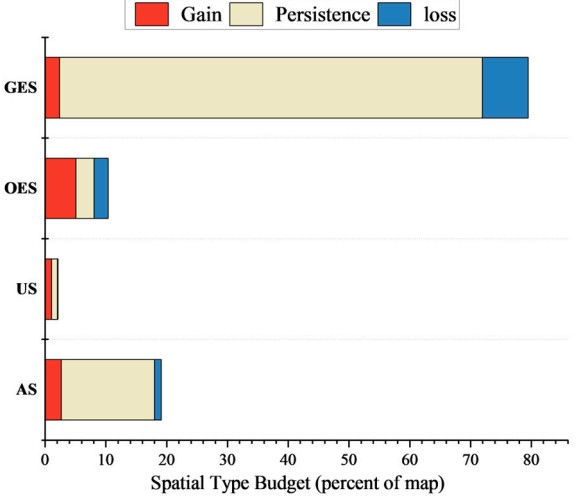

**Figure 5.** Gain, persistence, and loss in the TZS, 2000–2020.

### 3.3.2. Change Intensity Analysis for the "Three-Zone Space"

Figure 6 shows the intensity of gains and losses by each spatial type from 2000 to 2020. The left side shows that OES had the largest area in terms of annual gains while GES has the largest area in terms of annual losses. The right side of Figure 6 shows the annual transition intensity ($S_t$), and the red dotted line represents the uniform intensity line. If the change intensity of a certain spatial type is greater than the uniform line, it indicates that the spatial-type change is active; otherwise, it is dormant. OES was an active loser and active gainer, and OES's gain was greater than its loss, resulting in a net gain. In contrast, GES was a dormant loser and a dormant gainer, and GES's loss was greater than its gain, resulting in a net loss. In addition, US and AS were dormant in loss but active in terms of spatial gain.

Figure 7 shows the analysis results of transition intensities among the four spatial types from 2000 to 2020, where the red dashed line represents uniform transition intensi-

ties. If the transition intensity of a certain spatial type on the right side is greater than the uniform line, then the transition is targeted; conversely, it is avoided.

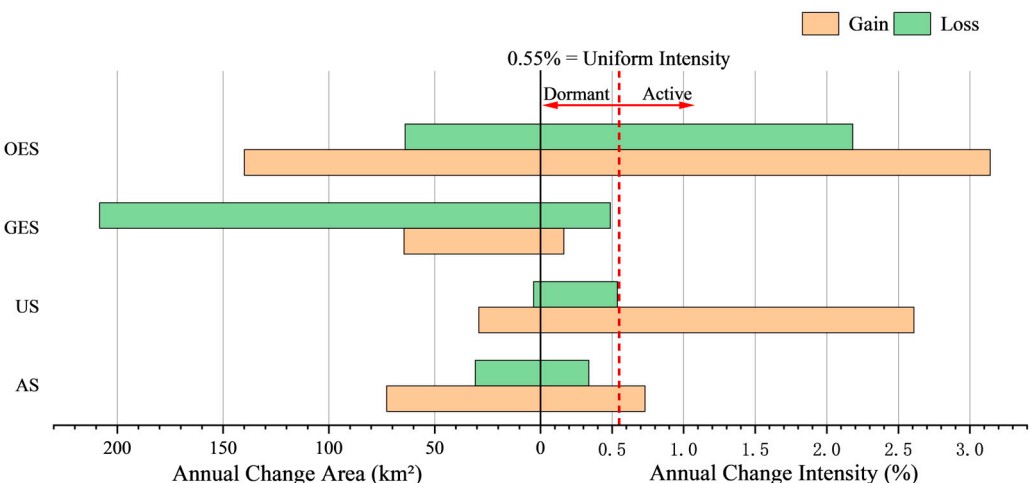

**Figure 6.** Annual change area and intensity of spatial categories' gains and losses in the IRB during 2000–2020.

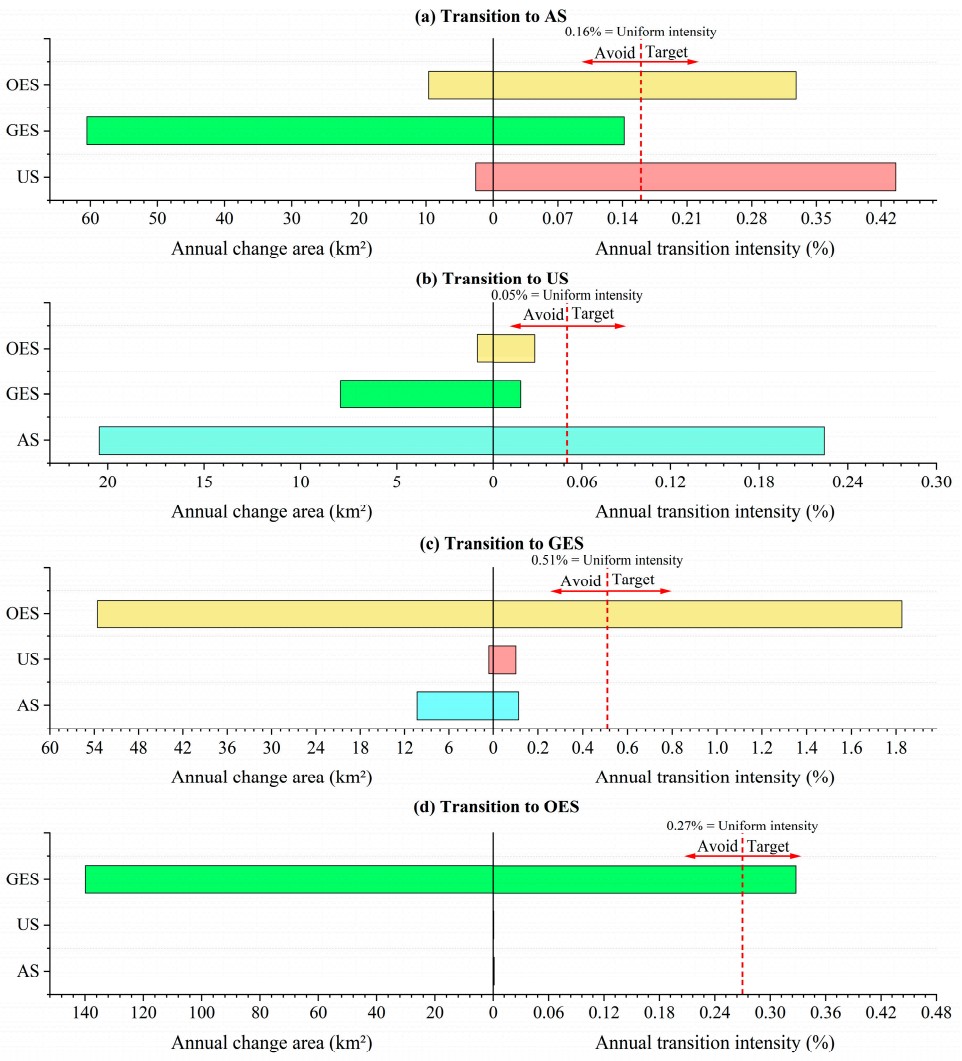

**Figure 7.** The gain intensity from losing spatial categories during 2000–2020.

The left side of Figure 7a shows that the largest source of increase for AS was from GES in terms of the annual change area. The right side of Figure 7a shows that the intensity of US transitioning to AS was the highest in terms of the annual transition intensity, followed by the transition of OES to AS, but the transition intensity of GES to AS was the lowest. This indicates that US and OES were more inclined to transition to AS, while GES avoided transitioning to AS. Further explanation is that, although the scale of GES transforming into AS was large, its transition intensity, because of its large initial area, was small and displayed a tendency to avoid.

The left side of Figure 7b shows that the main sources of increase for US were AS and OES in terms of the annual change area. The right side of Figure 7b shows that, in terms of the annual transition intensity, only AS targeted the transition to US specifically. Despite the large area of GES transitioning to US, its transition intensity remained low and displayed a tendency to avoid.

The left sides of Figure 7c,d show that, in terms of the annual change area, the main source of increase for both GES and OES was the other. The right sides of Figure 7c,d show that in terms of the annual transition intensity, GES and OES tended to transition to each other, but the transition intensity of OES to GES was stronger than that of GES to OES.

### 3.3.3. Spatial Differentiation Characteristics of Cross-Transitions in the "Three-Zone Space"

To explore the TZS conversion in the IRB, the TZS transformation matrix for 2000–2020 was obtained (Table 8) and visualized. Figure 8 shows the overall distribution of spatial changes during the period. In order to further study the distribution of various spatial type conversions, the kernel density analysis method was used (Figure 9).

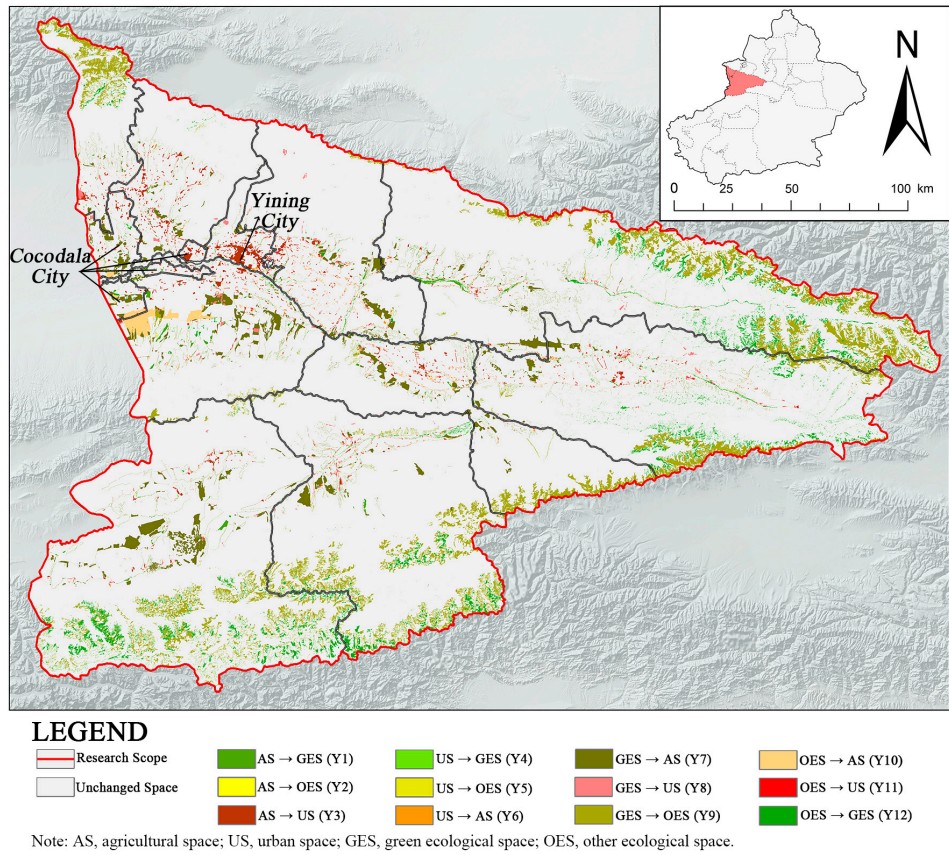

**Figure 8.** Cross-conversion map of three-zone space in the Ili River Basin from 2000 to 2020.

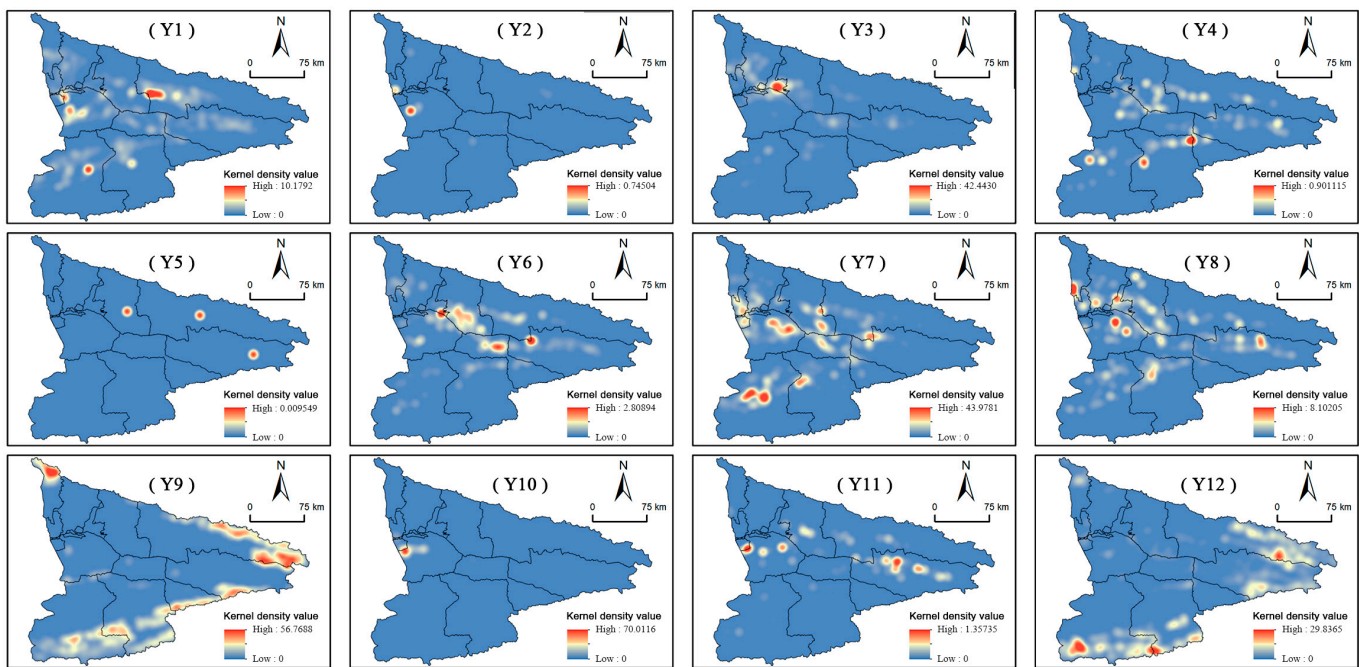

**Figure 9.** Spatial characteristics of the kernel density in the IRB for the conversion of the "three-zone space" from 2000 to 2020.

Figure 9 shows that the conversion from AS to GES (Figure 9(Y1)) in 2000–2020 exhibited pronounced spatial clustering characteristics. The high-value areas were mainly distributed in the eastern parts of Nileke County and Chabuchaer County. The spatial clustering of the conversion from AS to OES (Figure 9(Y2)) was high, and the areas with high values were distributed in the western parts of Chabuchaer County. The conversion from AS to US (Figure 9(Y3)) showed obvious spatial clustering characteristics, and the areas with high values were distributed in the southern part of Yining City and Cocadala City.

The high-value areas of the conversion from US to GES (Figure 9(Y4)) were widely distributed, and the spatial agglomeration was low. The high-value areas were mainly distributed in the central part of Nileke County, with secondary occurrences in western Tekes County and Zhaosu County. The spatial clustering of the conversion from US to other AS (Figure 9(Y6)) was high, and the areas with high values were distributed at the confluence of Yining City, Yining County, and Chabuchaer County, as well as the Gongliu County seat and the northwest of Xinyuan County. Notably, the conversion from US to OES (Figure 9(Y5)) from 2000 to 2020 only amounted to 0.05 km², which was limited to Yining County, Nileke County, and Xinyuan County and lacked a clustered pattern. Because of the sparse sample point data, further analysis of this type of spatial transformation was not conducted.

The GES conversions to AS (Figure 9(Y7)) and to US (Figure 9(Y8)) were mainly distributed in urban suburbs at all levels within the IRB. The spatial clustering of the conversion from GES to OES (Figure 9(Y9)) was high, the areas with high values were distributed in the Dzungarian Alatau range, the eastern regions of Borohoro Mountain, and the southern regions of Tian Shan Mountains in the IRB.

The spatial clustering of the conversion to AS (Figure 9(Y10)) was very high, and the areas with high kernel density values were distributed in the southern area of Cocadala City and the western boundary of Chabuchaer County. Concerning the conversion to US (Figure 9(Y11)), the areas with high values were distributed in the counties and cities along the Ili River and the Gongnaisi River. The spatial clustering of the conversion from OES to GES (Figure 9(Y12)) was also high. The analysis of Y9 and Y12 revealed a mutual conversion pattern between OES and GES, demonstrating significant spatial congruence between these two categories of spatial transformation.

*3.4. Driving Mechanism*

3.4.1. The Conversion of Agricultural Space

Regarding the conversion from US to AS, the order of the Q-statistic (value > 0.1) was as follows: X6 > X5 > X2 > X3 > X4 > X1 > X11 > X16 > X8 > X14 > X17 > X12 > X18 (Figure 10(Y6)). Therefore, the resident population and urbanization rate were the main factors for the conversion from US to AS. In this conversion process, the interaction among variables demonstrated bi-variable enhancement, nonlinear enhancement, and nonlinear weaken. Regarding the transformation from OES to AS (Figure 10(Y10)), the variables with an influence (Q > 0.2) were X18, X14, X12, X2, X11, X1, X3, X4, X17, X5, X6, and X16. In this transformation process, all variables exhibited bi-variable enhancement or nonlinear enhancement, indicating that the interaction effects among variables enhanced the explanatory power of the conversion from OES to AS to varying degrees. The interaction between X18 and X12 demonstrated bi-variable enhancement, with the highest interaction statistic reaching 0.87. Based on the location advantages of the newly built Cocodala City, it drove agricultural development in surrounding areas, thereby promoting the transformation from OES to AS.

Based on a comparison of the q values in all the transformations, the Q-statistic in the conversion from GES to AS (Figure 10(Y7)) was lower than others. The order of the Q-statistic (value > 0.1) was as follows: X14 > X8 > X1 > X16 > X6 > X17. Notably, the interaction between X15 and X16 demonstrated nonlinear enhancement, with an interaction statistic of 0.39. The influence of interactions in locational conditions and climatic factors surpassed the individual effects of each factor. The suitable climatic conditions for agricultural development, along with transportation facilities that facilitate people's agricultural activities, promoted the transformation of GES into AS.

3.4.2. The Conversion of Urban Space

Regarding the conversion from AS to US (Figure 10(Y3)), the variables with an influence (Q > 0.2) were X6, X5, X2, X4, X3, X18, X1, X8, X11, and X17. The interaction detection results for Y3 revealed that the interaction between X3 and X12 exhibited bi-variable enhancement, with the highest explanatory power reaching 0.7231. The rapid development of towns and cities, which occupied a certain amount of farmland, was the main reason for the transformation of AS into US.

The analysis of driving variables behind the transition from OES to US (Figure 10(Y11)) indicated that these spatial transformations were influenced by variables encompassing economic, social, geographical, locational, and climatic aspects. The order of the Q-statistic (Q > 0.2) was as follows: X1 > X4 > X6 > X3 > X5 > X14 > X12 > X11 > X18 > X16. The interaction detection results for Y11 revealed that all variables exhibited bi-variable enhancement or nonlinear enhancement. X14 and X12 demonstrated bi-variable enhancement, with the highest interaction statistic being 0.66.

In the analysis of the driving factors for the conversion from GES to US (Figure 10(Y8)), the variables with an influence (Q > 0.2) were X3, X6, X2, X5, X1, X4, and X16. This transformation process included bi-variable enhancement or nonlinear enhancement among all variables, indicating that the interaction effects among variables enhanced the explanatory power of these spatial transformations relative to single-variable effects. The interaction between X3 and X16 demonstrated the highest interaction statistic of 0.7645. Social and economic factors constituted the main driving forces. The development of urbanization caused cities to expand outward, inevitably encroaching on GES.

3.4.3. The Conversion of Ecological Space

The order of the Q-statistic (value > 0.1) corresponding to the transition from AS to GES (Figure 10(Y1)) was as follows: X18 > X14 > X11 > X8 > X1 > X17>X16> X4 > X5 > X12 > X13 > X16. The variables with an influence (Q > 0.2) were X18 and X14. The interaction detection results for Y1 revealed a pattern of bi-variable enhancement or nonlinear enhancement among the variables. This suggests that the interaction effects between vari-

ables enhanced the explanatory capability of the transition from AS to GES compared with single-factor effects, with differing degrees of strengthening. The largest interaction variables were X18 and X14, and the interaction statistic was 0.701. This indicates that both climatic factors and location conditions significantly impacted the spatial transformation of this type. The areas with certain water resources and relatively inconvenient transportation were more likely to undergo a transformation from AS to GES, which is closely related to the government's policies of returning farmland to forests and other forestry and grassland projects.

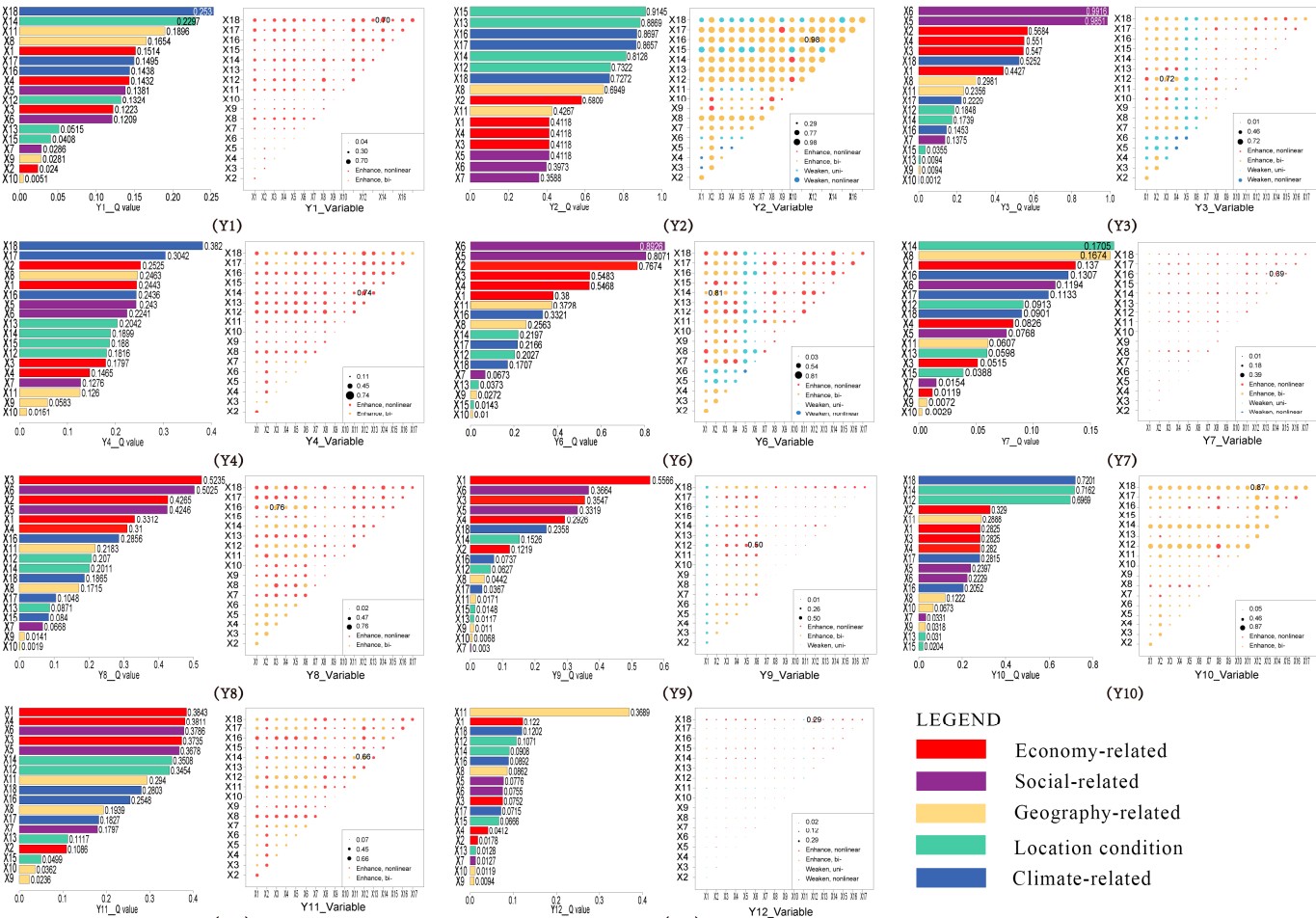

**Figure 10.** Geodetector results for the "three-zone space". Note: 1. In each group, the bar chart on the left shows the results of the factor detector, while the heat map on the right shows the results of the interactive detector. 2. Output value of primary production (X1), nighttime light index (X2), local general budget revenue (X3), local general budget expenditure (X4), resident population (X5), urbanization rate (X6), population density (X7), elevation (X8), slope (X9), slope aspect (X10), soil type (X11), the average distance to city and county seats (X12), average distance to rivers (X13), average distance to railways (X14), average distance to roads (X15), average annual temperature (X16), average annual precipitation (X17), and snow depth (X18). 3. Due to an inadequate number of sampling points in Y5, we are unable to fully leverage geographical detectors for a comprehensive exploration.

Regarding the transformation from US to GES (Figure 10(Y4)), the variables with an influence (Q > 0.2) were X18, X17, X2, X8, X1, X16, X5, X6, and X13. In this transformation process, the interaction among variables demonstrated bi-variable enhancement and nonlinear enhancement. The largest interaction variables were X14 and X12, and the interaction statistic was 0.74. Increasing public green spaces, constructing ecological parks, and

other means enhanced the urban ecological environment, leading to the transformation of US into GES.

The analysis of the driving factors for the conversion from OES to GES (Figure 10(Y12)) showed that the order of the Q-statistic (value > 0.1) was as follows: X11 > X1 > X18 > X12. The variables with an influence (Q > 0.2) were X11, with a q value of 0.3683. Among the interaction detection results, the interaction factors between X18 and X12 demonstrated nonlinear enhancement, with the highest value of interaction factors being 0.2937. This indicates that both snow depth and location conditions impacted the spatial transformation of this type.

The analysis of the driving factors behind the transformation of AS into OES (Figure 10(Y2)) revealed that, apart from slope (X9) and slope aspect (X10), the Q-statistic of other driving factors exceeded 0.2. Among them, the *Q* values of the X15, X16, X13, X1, X14, X12, X18, and X8 variables were greater than 0.6. The interaction between X16 and X12 yielded the highest explanatory power of 0.9841. Therefore, the distance from cities and towns and the climatic conditions were the main reasons for the abandonment of AS.

Regarding the conversion from GES to OES (Figure 10(Y9)), the variables with an influence (Q > 0.2) were X1, X6, X3, X5, X4 and X18. In this transformation process, except for the nonlinear weaken between X1 and the other variables, the other variables exhibited bivariable enhancement or nonlinear enhancement, suggesting that the interaction between X1 and the other factors did not enhance its explanatory power relative to a single variable. Within the interaction outcomes, the interaction factors between X6 and X12 displayed nonlinear-enhancement, with the maximum interaction variable value reaching 0.50.

## 4. Discussion

### 4.1. Social and Economic Factors Have a Significant Impact on the Evolution of Urban Space

Since 2000, the IRB has experienced rapid development through the pairing assist programs in XUAR [56], economic assistance, the Silk Road Economic Belt [57], and China's Western Development Strategy [58,59]. In terms of industrial structure changes, the growth rates of the primary, secondary, and tertiary industries have reached 515.59%, 1186.89%, and 2006.50%, respectively (Figure 11). The rapid development of the tertiary industry also resulted in a shift in its proportion from 29.2% in 2000 to 50.4% in 2020, compared with the primary and secondary industries, which changed from 45.7% and 35.4% to 23.1% and 26.5%, respectively. This shift aligns with the increase in US within the IRB. Under the background of urbanization and industrialization, the territorial space pattern of the IRB underwent changes.

Since 2000, US expanded significantly. US was primarily distributed along the banks of the Ili River, Kash River, Tekes River, and other rivers in the valley lowland, with urban sprawl expanding outward from the city center into surrounding agricultural areas and intersecting with them, causing the patch shape to become complex and the level of space fragmentation to become larger (Figure 3). The conversion of US was primarily sourced from AS (0.74%), followed by GES (0.29%) and OES (0.03%). The factors influencing their conversions were predominantly human factors and exhibited significant spatial heterogeneity. In the case of AS→US (Y3), social and economic factors played the predominant driving roles, with the urbanization rate and resident population as the main driving factors. As for the conversion of GES→US (Y8) and OES→US (Y11), social and economic factors also made the greatest contribution. Compared with ecological space, AS incurred lower construction costs and was more readily converted to US. The expansion of US was concentrated in Khorgos City, Cocodala City, and Yining City. As the capital of the autonomous prefecture, Yining City serves as a regional transportation hub, with a population of 778,000 residents as of 2020, boasting relatively high levels of fiscal revenue, expenditure, and urbanization rates. This signifies that elevated levels of economic development can lead to greater conversion capabilities. Cocodala City, established in 2015 and positioned as a part of the XUAR border urban belt, is located between Khorgos City and Yining City. Its urban development has been bolstered by support from both national and

local strategies. Because of urban construction needs, Cocodala City boasts the highest general fiscal revenue and expenditure in the region. The acceleration of urban construction and economic investment has promoted the expansion of US. Khorgos City, home to Khorgos Port, is one of the largest land ports in Central Asia. It has experienced rapid growth in border trade. In 2020, the import and export cargo volume at Khorgos Port reached 34.4162 million tons, ranking first among XUAR ports, with an import and export trade value of CNY 242.65 billion. This enhanced its status as a land port city, attracting large flows of people, goods, and information, thus driving US expansion.

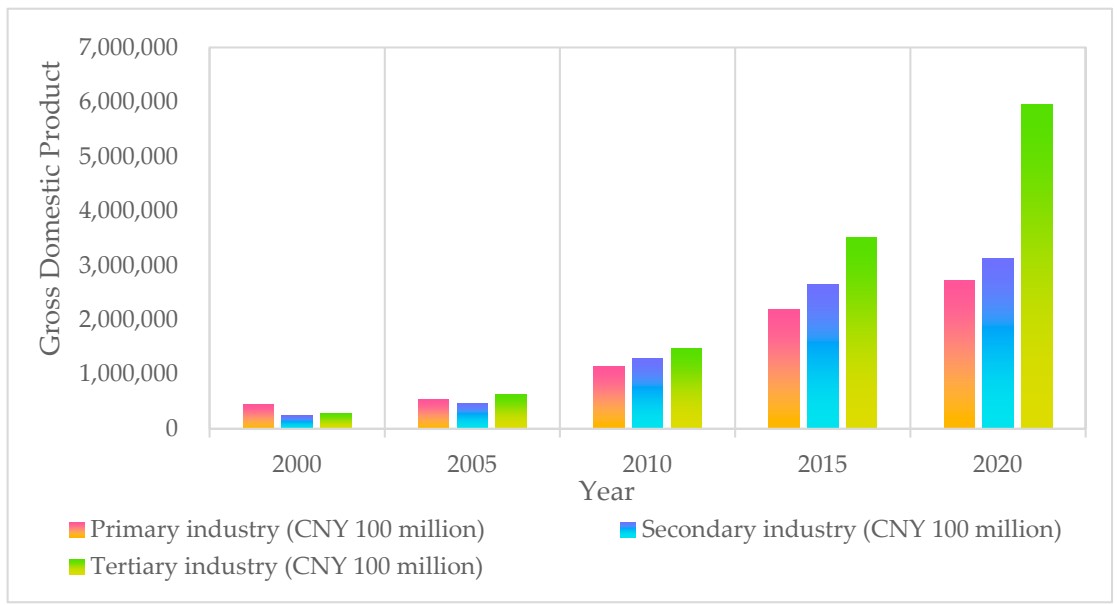

**Figure 11.** Trend of industrial changes in the Ili River Basin from 2000 to 2020.

*4.2. The Scale of Agricultural Space Did Not Change Significantly*

AS increased from 9,123.57 km$^2$ in 2000 to 9,961.07 km$^2$ in 2020. On the one hand, 408.86 km$^2$ of AS was converted to US. On the other hand, 1209.95 km$^2$ of GES, 192.51 km$^2$ of OES, and 52.25 km$^2$ of US were used to supplement AS. Therefore, the overall AS remained stable with consistent growth within the IRB. This is different from the situation in the adjacent area in Almaty province, Kazakhstan, where there was a decrease in cultivated land and an increase in grassland [60]. The Chinese government has implemented a "requisition–compensation balance" policy of cultivated land [61]. The government has resolutely stopped the "non-agricultural" use of cultivated land and implemented a high-standard cultivated land construction action plan to improve the quality and productivity of cultivated land.

The conversion of AS was primarily sourced from GES (2.19%), followed by OES (0.35%) and US (0.09%). The factors influencing their conversions were predominantly natural factors. In the case of GES→AS (Y7) and OES→AS (Y10), locational factors and geographical factors played a predominant driving role. This spatial transformation entails a comprehensive assessment of transportation, arable land suitability, and snow depths in OES, fulfilling the requirements for agricultural development. The presence of convenient transportation and a favorable climate accelerates agricultural development, facilitating the transformation of GES and OES into AS. Regarding the conversion of US, social and economic factors made the highest contribution. The scale of the transition from US to AS (Y6) was relatively small between 2000 and 2020. Social and economic factors such as the permanent population, urbanization rate, and local general budget revenue and expenditure significantly influenced this transition. Polarization and redistribution of the population in the region and rapid urbanization led to the degradation of inefficient land use in some peripheral towns and villages into AS, which optimized the structure of AS.

### 4.3. Human and Natural Factors Led to the Degradation of Ecological Space

The IRB is situated in the western Tian Shan Mountains and is surrounded by mountains on three sides. It features prevalent westerly winds and abundant precipitation, making it a "wet island" in Central Asia, though it remains part of the arid inland region. Global climate change has impacted the IRB's ecological environment, making it an ecologically fragile area and an important ecological functional zone in China. Under the influence of the policies of returning farmland to forest (grass), returning grazing land to grasslands, natural forest resources protection, and ecological civilization construction, ecological issues have been emphasized. Our research found that from 2000 to 2020, OES in the IRB, mainly bareland, increased by 1518.82 km$^2$, while GES decreased by 2875.97 km$^2$.

The conversion of GES is primarily sourced from OES (1.94%), followed by AS (0.37%). The transformation from AS to GES (Y1) tended to occur in remote areas with far from towns because of transportation inconveniences. This conversion was mainly concentrated in Nileke County and Chabuchaer County. Nileke County is surrounded by high mountains, has intensified its efforts to protect the ecological environment, including land, forests, and water systems, and is actively creating a practical innovation base for the concept that "lucid waters and lush mountains are invaluable assets." Through measures such as afforestation of barren mountains and environmental management, it has promoted the improvement and enhancement of the ecological environment. In Chabuchaer County, conversions from AS to GES are closely tied to the government's ongoing efforts to promote the protection and restoration of natural forests, as well as the implementation of grassland restoration projects, such as the "returning grazing land to grasslands" and "returning farmland to forests and grasslands" policies.

The conversion of OES was primarily sourced from GES (5.05%). In the conversion from GES to OES, economic, social, and climatic factors played major driving roles. On the one hand, the encroachment of other spaces on GES led to the degradation of green spaces. On the other hand, the conversion of GES to OES often occurred in mountainous areas, where various geological activities and climate changes caused glaciers and perennial snow to melt, forming bareland. Regarding the conversions from OES to US (Y11), climatic and locational factors played the predominant role. Such transformations typically occurred in areas abundant in water resources at the periphery of cities, closely intertwined with the implementation of regional ecological conservation projects. This was mainly concentrated in Cocodala City and Xinyuan County, which are expanding their urbanization processes by utilizing OES. In conversions from OES to GES (Y12), geographical and locational factors played dominant roles, which were distributed in high-altitude areas at around 3000 m.

Notably, the mutual transformation between GES and OES was significant and demonstrated a clear spatial congruence in these two categories of spatial transformation (Figure 9(Y9,Y12)). This transformation was primarily distributed in the southern region of the Tian Shan Mountains, the Borohoro range, and the Dzungarian Alatau range, overlapping with the distribution of glaciers and permanent snow cover. The reduction in glacier and permanent snow cover areas led to an increase in OES (Figure 2). Glaciers, as "solid reservoirs," are crucial sources of replenishment for many rivers in arid regions, playing a vital role in maintaining regional ecological stability and regulating river runoff. The existing research indicates that the snowline elevation in various regions of the Tian Shan Mountains showed a fluctuating upward trend from 2001 to 2015. The average snowline elevation in the Tian Shan Mountains is 3690 m, with a significant upward trend rate of 276 m per decade. Meanwhile, the snowline in the IRB is 3390 m, with a trend rate of 180 m per decade [62]. The meteorological factors most closely related to glacier changes are temperature and precipitation. In recent decades, XUAR has experienced rising temperatures and increased precipitation, which have profoundly impacted glacier reserves. The rise in temperature accelerates glacier melting, which partially offsets the replenishment of glaciers from increased precipitation, promoting glacier melting. Consequently, the increased meltwater is a rising trend in the runoff of glacier-fed rivers in the Tian Shan region

of XUAR [63]. This trend also explains the increase in water bodies and wetland areas in the IRB (Figure 2). The conversion of OES to GES in mountainous regions resulted from the natural growth and evolution of bareland vegetation, driven by climate warming and increased rainfall over the 20-year period. Vegetation indices vary with different altitudes. According to the existing research in the IRB, vegetation in areas with elevations below 500 m, between 500 and 1000 m, and above 3000 m has shown an improving trend, while vegetation in areas with elevations between 1000 and 3000 m has experienced vegetation degradation [64]. Consequently, the rise in the snowline and glacier melting leads to the emergence of bareland, which forms new OES through natural restoration processes. This indicates that the IRB should place greater emphasis on managing ecological space and effectively controlling the reduction in ecological space, which will lead to the future development of the social economy in a greener and higher-quality direction.

### 4.4. Research Significance and Limitations

To understand the inherent mechanisms of spatial evolution and interactions among spaces in the region, an in-depth interactive study from the TZS perspective was conducted. The findings provide a basis for the governance and protection of the IRB, offering guidance for sustainable land management. This paper selected variables from five aspects, including economy, society, geography, location conditions, and climate, to analyze the driving mechanisms of the TZS in the IRB. However, to form a complete theoretical framework of driving mechanisms, further exploration is still required. Because data are difficult to obtain, only district- and county-level indicators were selected. The impact factors also have shortcomings. Subsequent studies can consider indicators such as policy systems, legal regulations, and other indicators, improve the selection of indicators, and fully explore the factors affecting the development of the TZS.

### 5. Conclusions

The ecological environment of the IRB is of great significance to the ecological security of Northwest China and Central Asia. Based on the territorial spatial governance framework of the TZS, this paper analyzes the long-term spatiotemporal evolution characteristics of the TZS in the IRB and examines the inherent driving mechanisms of its evolution. The following conclusions are drawn from this study.

First, from 2000 to 2020, the AS in the IRB experienced a net increase of 837.51 km$^2$, with a dynamic degree of 0.92%, indicating a gradual expansion trend. US and OES expanded by 519.64 km$^2$ and 1518.83 km$^2$, with the dynamic degrees of 8.67% and 5.17%, respectively, showing significant expansion trends in these two spatial categories. Conversely, GES reduced by 2875.97 km$^2$, with the dynamic degree recorded at $-0.67$%, signaling a declining trend. Taking 2010 as a pivotal year, the trends manifested as a "decline–rise" in AS, "rise–rise" in US, "rise–decline" in OES, and "decline–decline" in GES.

Second, from intensity analysis, the total TZS change in IRB was 11.07%. At the spatial-type level, both the increase and decrease intensities of OES were active, while those of GES were dormant. The increased intensities of US and AS were active, but their decreased intensities were dormant. The increase in AS mainly came from US and OES, while the increase in US was mainly sourced from AS. The main source of increase for both GES and OES was each other. In terms of spatial transformation intensity, US and OES tended to transform into AS; AS tended to transform into US; and OES and GES had a mutual transformation tendency.

Third, during the 20 years, 12 types of space transformations occurred within the study area. AS converted into US, especially around emerging cities like Khorgas and Cocodala. Additionally, the conversion towards GES was scattered, which encompassed a wider spatial range. The mutual conversion between OES and GES showed spatial distribution consistency, mainly occurring in the Borohoro ranges and the Halik ranges.

Finally, regarding the driving mechanisms, the evolution of US in the IRB was driven by social and economic factors. Location and climate factors accelerated agricultural de-

velopment, facilitating the transformation of GES and OES into AS. Climate and economic factors played a crucial role in the scale of conversions between OES and GES.

**Author Contributions:** Conceptualization, M.Y.; methodology, Z.J. and M.Y.; software and visualization, Z.J. and L.Y.; validation, Z.J. and M.Y.; formal analysis, Z.J.; investigation, Z.J., W.S. and L.Y.; resources, Z.J.; data curation, Z.J., W.S. and Z.L.; writing—original draft preparation, Z.J. and M.Y.; writing—review and editing, M.Y.; supervision, Z.J. and M.Y.; project administration, M.Y.; funding acquisition, M.Y. All authors have read and agreed to the published version of the manuscript.

**Funding:** This research was funded by the National Natural Science Foundation of China (Grant No. 82260414) and the Social Science Fund Project of XPCC (Grant No. 21YB12).

**Data Availability Statement:** Data are contained within the article.

**Acknowledgments:** We are sincerely grateful for the editors and reviewers who commented on this paper and gave their time and effort. Finally, we would like to thank the National Natural Science Foundation of China and the Social Science Fund Project of XPCC for their support of this study.

**Conflicts of Interest:** The authors declare no conflicts of interest.

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
