# Peer review of "Spatial–Temporal Evolution Characteristics and Driving Mechanism Analysis of the “Three-Zone Space” in China’s Ili River Basin"

_land, doi:10.3390/land13091530_

Round 1

Reviewer 1 Report

Comments and Suggestions for Authors

This manuscript focuses on the understanding the spatiotemporal changes and driving mechanisms of the “three zone space” (TZS) in the IRB which is of significant practical importance for promoting sustainable development and optimizing the territorial spatial pattern. Although findings in the manuscript are interesting and valuable for scientific guidance for sustainable land management, it still needs major revision and check before formal publication. Here are some specific suggestions for the authors:

1.       At Line 400: The loss of US was largely converted to AS and vice versa.

At Line 554: 52.25 km2 of US were used for supplementing the AS.

At Line 577: As for the conversion of US, social and economic factors made the highest contribution.

About the three sentences above, it is recommended that where it is happened as to the “US converted into AS” and what reasons makes this happened. Authors mentioned one of the reasons is the implementation of a requisition-compensation balancepolicy of cultivated land at Line 559. However, it is better for authors to tell us where is required for urban space from cultivated land and where is compensed for the cultivated land from the urban space.

 2. As for the “Dynamic degree” and “Space Patterns Transfer Matrix” in the Section 2.2.2 and Section 2.2.4:

Recently some studies have shown that the two indicators of the dynamic degree model have limitations. The “Intensity Analysis” method can compensate these shortcomings. So it is recommended that the authors consider this method, which helps get new insights about the land change process of study area. What is more, the methods summary three change components, i.e. Quantity, Exchange and Shift and could naturally appear some figures rather than description in words so that the change analysis is more in-depth. The authors can refer to the following sources:

(1) Aldwaik S Z, Pontius Jr R G. Intensity analysis to unify measurements of size and stationarity of land changes by interval, category, and transition [J]. Landscape and Urban Planning, 2012, 106(1): 103-114.

(2) Feng Y, Lei Z, Tong X, et al. Spatially-explicit modeling and intensity analysis of China's land use change 2000-2050 [J]. Journal of Environment Management, 2020, 263: 110407.

(3) Pontius Jr R G, Huang J, Jiang W, et al. Rules to write mathematics to clarify metrics such as the land use dynamic degrees [J]. Landscape Ecology, 2017, 32(12): 2249-2260.

(4) Pontius Jr R G, et al. Design and Interpretation of Intensity Analysis Illustrated by Land Change in Central Kalimantan, Indonesia[J]. Land, 2013, 2: 351-369.

3. Line 106 and 107: “...... ascertain whether land use transitions are systematic or random.” Authors could clearly tell us or explain the concept about the systematic or random in the Methods Section.

 4. Line 165: Authors need give the accuracy of the GlobeLand30 used in this research area.

 5. Line 187 about the 2.2.3. Kernel density estimation: How to perform the KDE? What kind of the software of GIS is conducted for this operation of the KDE? How many meters is the search radius h in this study?

 6. It seems that the vertical axis in Figure 7 does not follow standard conventions. So it is recommended that the authors might add a description on the left side of the vertical axis in Figure 7 to enhance clarity and completeness.

 7. As for the Conclusions Section. If it is possible, please pay attention to the conciseness in the Conclusions Section.

Comments on the Quality of English Language

Good

Reviewer 2 Report

Comments and Suggestions for Authors

I found the manuscript interesting and well organized and I appreciate the authors nice work. However, I have comments that I think need attention and some of them may have major impacts on the paper:

1- I could not open the link given for Data GlobeLand30 in Table 1. The other link (www.webmap.cn) was working but it was all in Chinese and almost impossible to navigate for a non-Chinese person. And I guess all variables in Table 4 are derived from these sources, but some variables (such as soil type) can not be found directly in the list of Table 1 and it is better that all variables in Table 4 are linked to a data source in Table 1.

2- Explanation given for kernel density estimation (lines 192-195) is a bit ambiguous. What do you mean by "TZS transition to and from the surface"? What is the x variable?

3- I guess equation#6 should be written as Max(...) + Min(...).

4- In lines 238-242, you state that Dij around zero means random inter-category transition and large positive or negative values means systematic transitions. So my understanding is that with 'systematic transitions' you mean that those transitions have specific reasons and it make sense that those transitions be associated with some variables, is it right? If yes, I suggest to elaborate on this concept a bit more to help reader understand the concept better. Then, you state in lines 245-247 that for Rij the more positive value indicate a systematic transition and a more negative value means avoiding a systematic transition. Given that Rij is just normalized Dij, I expect that for Rij the interpretation would also be the same as Dij but it is not the same. It would be good to elaborate more on this to help reader understand the concepts better.

5- It would be nice to inform the reader a bit more on the science behind OPGD package when introducing the formula (9) in lines 266-268. What is H? You say 'number of units' but it is ambiguous in your context. It would also be great if you can provide formula for calculating Q for interaction, but this would be optional.

6- The indicators given in Table 4 are all provided or transformed to a spatial map with the same resolution and projection, right? Could you please provide a bit more information about this spatial processing?

7- I could not find any explanation or usage of figure 4.b (chord diagram) in the text, please explain it in the text. Also, does the exact value of density numbers given in section 3.3.1 (e.g. the figure of 9.58 in line#349) has a specific meaning? I think the point in kernel analysis is to reveal a spatial pattern, and the density values are not comparable directly. I think the big circles of figure 5-Y5 also deserve some discussion.

8- This is an important thing: I did the calculations for table 9 using the formulas provided in the manuscript but was not able to get the results given in that table. I have attached the excel sheet of my calculations and I appreciate if you check it and let me know what is wrong with it.

9-The factor analysis in section 3.4 is in-depth and covers a lot of relations. The only problem is that the reader can not get a clear picture of how the variables come into play and interact with each other out of these graphs. For each of 12 spatial types, it would be better to give a sentence to justify WHY the top variables selected by Q statistics are important for that spatial type, and why the interaction is significant or not significant. After knowing being weakening or enhancing, further details on the interaction (e.g. linear or non-linear, etc.) does not seem to be used anywhere in your analysis and this issue may be simplified. Also, connection of this section to the discussion section should be checked again. For example, in line#531 it is said that variables X5 and X6 are the main driving factors in AS conversion, but when I look at the section 3.4.2, I see other variables such as X3 and X1 shown as the most important ones.

Regards

Round 2

Reviewer 1 Report

Comments and Suggestions for Authors

Authors have already revised the manuscript according to my suggestions. So I agree. 

Comments on the Quality of English Language

Good

Author Response

Thank you for your positive review and support. We are delighted that you find our work suitable for publication.

Reviewer 2 Report

Comments and Suggestions for Authors

I read the revised manuscript and I appreciate the time and effort the authors spent on its major revision. I think the manuscript is acceptable for publication, but I think below points need to be addressed before that:

1- I think in all of the formulas (4)-(10), J should be replaced by n. In formula 3, the n is defined to be the number of categories and J serves the same thing but has not been defined elsewhere. Also, you have defined Cj+ and C+j in lines 188-189 but they are never used.

2- I think formula (11) is related to one variable (v) and not the interaction of two variables and M might be the number of strata for that variables (resulting from quantization as noted in the manuscript). Therefore, this phrase in line#234 is confusing: "M represents the explanatory variables or strata of explanatory variables". Besides, the stratification itself is ambiguous. Do you mean the 3 to 8 categories you mentioned in line#229?

3- Although trivial, I suggest to mention that annual values (and intensity threshold) mentioned in figures 6 and 7 are calculated by dividing the total amount over 2000-2020 by the number of years (20).

Thanks for your contribution to the field knowledge.
